# Lung Ultrasound Findings in Healthy Children and in Those Who Had Recent, Not Severe COVID-19 Infection

**DOI:** 10.3390/jcm11205999

**Published:** 2022-10-11

**Authors:** Massimiliano Cantinotti, Pietro Marchese, Nadia Assanta, Alessandra Pizzuto, Giulia Corana, Giuseppe Santoro, Eliana Franchi, Cecilia Viacava, Jef Van den Eynde, Shelby Kutty, Luna Gargani, Raffaele Giordano

**Affiliations:** 1Fondazione G. Monasterio CNR-Regione Toscana, 54100 Massa, Italy; 2Institute of Clinical Physiology, 56127 Pisa, Italy; 3Institute of Life Sciences, Scuola Superiore Sant’Anna, 56127 Pisa, Italy; 4Department of Cardiovascular Sciences, KU Leuven, 3010 Leuven, Belgium; 5Helen B. Taussig Heart Center, Department of Pediatrics, Johns Hopkins Hospital, Baltimore, MD 21205, USA; 6Cardiothoracic Department, University of Pisa, 56127 Pisa, Italy; 7Adult and Pediatric Cardiac Surgery, Department Advanced Biomedical Sciences, University of Naples “Federico II”, 80131 Naples, Italy

**Keywords:** lung ultrasound, COVID-19, normal LUS findings, children

## Abstract

Background: Lung ultrasound (LUS) is gaining consensus as a non-invasive diagnostic imaging method for the evaluation of pulmonary disease in children. Aim: To clarify what type of artifacts (e.g., B-lines, pleural irregularity) can be defined normal LUS findings in children and to evaluate the differences in children who did not experience COVID-19 and in those with recent, not severe, previous COVID-19. Methods: LUS was performed according to standardized protocols. Different patterns of normality were defined: pattern 1: no plural irregularity and no B-lines; pattern 2: only mild basal posterior plural irregularity and no B-lines; pattern 3: mild posterior basal/para-spine/apical pleural irregularity and no B-lines; pattern 4: like pattern 3 plus rare B-lines; pattern 5: mild, diffuse short subpleural vertical artifacts and rare B-lines; pattern 6: mild, diffuse short subpleural vertical artifacts and limited B-lines; pattern 7: like pattern 6 plus minimal subpleural atelectasis. Coalescent B-lines, consolidations, or effusion were considered pathological. Results: Overall, 459 healthy children were prospectively recruited (mean age 10.564 ± 3.839 years). Children were divided into two groups: group 1 (*n* = 336), those who had not had COVID-19 infection, and group 2 (*n* = 123), those who experienced COVID-19 infection. Children with previous COVID-19 had higher values of LUS score than those who had not (*p* = 0.0002). Children with asymptomatic COVID-19 had similar LUS score as those who did not have infections (*p* > 0.05), while those who had symptoms showed higher LUS score than those who had not shown symptoms (*p* = 0.0228). Conclusions: We report the pattern of normality for LUS examination in children. We also showed that otherwise healthy children who recovered from COVID-19 and even those who were mildly symptomatic had more “physiological” artifacts at LUS examinations.

## 1. Introduction

Lung ultrasound (LUS) is widely implemented as a non-invasive diagnostic imaging method for the evaluation of pulmonary disease in children [1,2,3,4,5] and particularly in pediatric emergency departments and multiple pediatric fields for the screening, diagnosis, and follow-up of pulmonary disease [1,2,3,4]. Despite its rising use, systems to classify pulmonary disease severity by LUS in children are lacking, and even the definition of “normal LUS findings” remains unclear. It has been observed that even in children with healthy lungs, some artifacts such as rare B-lines and short subpleural vertical artifacts may be present [1,2]. These findings further complicate the differentiation between healthy and mildly pathological lungs on ultrasound assessment. Additionally, with the introduction of the COVID-19 pandemic, multiple novel questions have developed in this context [6,7,8]. These questions include whether COVID-19 infection leaves prolonged damage to the lungs. Further, if the former is the case, does this pulmonary damage relate to the severity of the disease? Many articles have focused on the role of LUS for the early diagnosis and follow-up of symptomatic COVID-19 in both adults [6] and children [9,10,11,12]. However, studies on the use of LUS evaluation in mid-term follow-up on asymptomatic pediatric COVID-19 patients are sparse.

The primary aim of this study is to identify which artifacts (e.g., B-lines, pleural irregularity) on LUS assessment in children can be defined as “normal lung ultrasound findings”. Secondarily, we aim to evaluate the differences in children without a history of COVID-19 infection compared to children with a history of previous recent, not severe COVID-19 infection (resolved by a minimum of 2 weeks) who were either asymptomatic or with mild-to-moderate respiratory symptoms.

## 2. Methods

In the present study, Caucasian children were prospectively recruited from healthy subjects presenting for evaluation at the congenital heart-defect screening program in the outpatient pediatric cardiology department at the Fondazione CNR-Regione Toscana G. Monasterio of Massa between April 2020 and March 2022. Neonates and infants were most often evaluated for “innocent murmurs”, while older children and young adolescents were usually referred to our outpatient department after physical evaluation in the context of sports screening with suspicions about existence of a heart defect.

Exclusion criteria included [12]: evidence of congenital or acquired heart disease, known or suspected neuromuscular disease, genetic syndromes, chromosomal abnormalities, pulmonary hypertension, systemic hypertension, connective tissue disease, or family history of genetic cardiac disease.

Children with recent pneumonia or other viral infection were included if the infection had resolved by at least 2 weeks and if they were completely asymptomatic at the moment of examination. Patients with a recent COVID-19 infection were included when negativization (documented by nasal swab) occurred by a minimum of 2 weeks. Previous COVID-19 symptoms were graded as none, mild, or moderate according to international classifications [13,14]. Children with severe infections were too limited and thus were excluded from final analysis. Children with COVID-19 infection from more than two months were also excluded since evidence from adults shows how radiological alterations are after two months in mild-to-moderate infection [15].

The study was approved by the local ethics committee (CE 62/2016). Parents or legal guardians were informed and agreed to participate by providing written consent.

Images were collected only in quiet and cooperative children. Infants could bottle feed, whereas older children could watch cartoons during examinations. No children were sedated during the assessments.

### Lung Ultrasound Examination

LUS examinations were performed by two experienced pediatric cardiologists (M.P., E.F.), and images were digitally stored in a central core lab. LUS was performed according to standardized protocols [2,5]. For each hemithorax, three major areas (anterior, lateral, and posterior) delineated by the para-sternal, anterior axillary, and posterior axillary line were identified. Every area was further divided into the upper and lower half, creating six different quadrants for each hemithorax: anterior superior, anterior inferior, lateral superior, lateral inferior, posterior superior, and posterior inferior [2,5]. Stored images were reviewed by an expert operator blinded of clinical examination (M.C.). B-lines, defined as comet-like artifacts, were analyzed. These are the ultrasound equivalent to Kerley B-lines, indicating subpleural interstitial edema [1,2].

The cut-off score for pathological findings was defined as 1, based on findings from one of our previous studies [14]. In this score system, a score of 0 is assigned to observations without significant B-lines (maximum 0–2 for each segment), a score of 1 to observations where significant separated B-lines can be detected, a score of 2 when significant coalescent B-lines are observed, and, finally, a score 3 is assigned to sites with complete loss of aeration (lung consolidation). Therefore, all patterns of normality are required to have a score below 1. Even children with pattern of normal LUS examination (e.g., score 0), however, often present a range of minor abnormalities (subpleural artifacts, B-lines), that may be present in isolation or in different scanning areas. Thus, we implemented the previous scoring system [14] by describing different patterns of minor abnormalities and arbitrary sub-classing the score of 0 into seven major patterns (Figure 1).

Pattern 1: Absence of sub-pleural artifacts and no B-lines, point 0;Pattern 2: Only mild basal posterior sub-pleural artifacts and no B-lines, point 0–0.1’;Pattern 3: Mild posterior basal, para-spine, and apical short subpleural vertical artifacts and no B-lines, point 0.1–0.2;Pattern 4: Mild posterior basal, para-spine, and apical short subpleural vertical artifacts and rare B-lines (e.g., ≤2 in each segment), point 0.2–0.4;Pattern 5: Mild, diffuse short subpleural vertical artifacts and rare B-lines (e.g., ≤2 in each segment), point 0.4–0.6;Pattern 6: Mild, diffuse short subpleural vertical artifacts and limited B-lines (e.g., ≤4 in each segment), point 0.6–0.8;Pattern 7: Mild, diffuse short subpleural vertical artifacts with mild subpleural areas of atelectasis and limited B-lines (e.g., ≤4 in each segment), point 0.8–1.

Examinations were excluded from analysis at presence of a poor acoustic window (e.g., not allowing a precise estimation of B-lines) and/or at incompleteness (e.g., images not available for at least two-thirds of the sites).

On observation of multiple B-lines for each segment (e.g., >4 in each segment), coalescent B-lines, consolidations, or effusion, the examination was, per definition, considered pathological [1,2].

In addition, rates of intra-observer and inter-observer variability were calculated from 20 randomly selected subjects (Figure 1 and Appendix A).

## 3. Statistical Analysis

All continuous variables were expressed as mean and standard deviation (SD). Categorical variables were expressed as frequency (%) and were compared with the chi-square test. Comparison between groups was performed using Student’s *t*-test or ANOVA test when appropriate. Patients were divided into four age groups and were subsequently evaluated: Age group 1 (31 days–≤24 months); age group 2 (2–≤5 years); age group 3 (5–≤11 years); and age group 4 (11–≤18 years) [14]. When age groups were compared with an unequal number of patients, a subgroup of patients from the larger group was randomly selected for 1:1 matching by BSA. Intra-and inter-rater reliability were assessed using intra-class correlation coefficients (ICC). Intra-rater reliability was defined as the degree of agreement among repeated measurements on the same scan performed by a single rater, while inter-rater reliability was defined as the extent to which two independent raters agreed. The coefficient of variation (CV) was calculated as an average value from individual CVs for all the duplicates. Inter-rater CVs below 15% are acceptable, whereas intra-rater CVs should be less than 10%. All analyses were performed using R Statistical Software (version 4.0.5, Foundation for Statistical Computing, Vienna, Austria). A two-tailed *p*-value < 0.05 was considered statistically significant.

## 4. Results

From April 2020 to March 2022, 475 subjects were prospectively recruited. Sixteen children were excluded due to either a positive anamnesis for a recent airway infection (*n* = 9) or the simultaneous use of bronchodilator therapy for allergic asthma (*n* = 4) or severe COVID-19 infection (*n* = 3). Thus, final analysis was conducted on 459 patients (male: 252 patients; female: 207 patients), with a mean age of 10.564 ± 3.839 years (range 0.293 days–17.868 years). Median BSA was 1.216 m^2^ (range 0.339–2.051 m^2^) (Table 1 and Figure 2).

Included children were divided into two major groups: group 1, patients with absence of COVID-19 infection, and group 2, patients with a COVID-19 infection. No significant differences in age, body weight, height, or body surface area (BSA) were observed between the two groups (*p* > 0.05).

### 4.1. LUS Score in Healthy Children Who Did Not Experience COVID-19 Infection

Group 1 (absence of COVID-19 infection) included 336 children with a mean age of 10.498 ± 3.941 years (range 0.293–17.868 years). On LUS assessment of group 1, mean global LUS score was 0.259 ±0.154, and left lung score was 0.262 ± 0.168, while mean right lung score was 0.254 ± 0.16 (*p* = 0.538). The most common observed pattern in this patient group was pattern 3 (50.89%). Pattern 3 was the most common observed pattern for both right (51.79%) and left lung (45.54%). After BSA matching, none of the patients in this group presented with pattern 7 (0%).

### 4.2. LUS Score in Healthy Children Who Had Previous COVID-19 Infection

Group 2 (presence of COVID-19 infection) included 123 children with a mean age of 10.706 ± 3.573 years (range 0.934–17.847 years). Children in group 2 had previous COVID infection at a mean of 28 ± 22 days before examination (range 16–60 days). No significant correlation between COVID-19 infection timing and LUS score were described (*p* all >0.05)

On LUS assessment of group 2, mean global LUS score was 0.306 ± 0.185, and mean left lung score was 0.306 ± 0.2, while mean right lung score was 0.308 ± 0.206.

The most common observed pattern in this group differed between right and left lungs. Pattern 4 was most frequently observed in the right lung (35.77%), whereas pattern 3 occurred most often in the left lung (39.02%). In contrast to group 1, some patients from group 2 were described with pattern 7. This occurred four times in the right lung (3.25%) and three times in the left lung (2.44%) (Table 2 and Table A1).

### 4.3. Comparison of Patients with Positive and Negative Anamnesis for COVID-19

Overall, patients with a demonstrated positive anamnesis for COVID-19 had a higher value of total LUS score than patients with a negative anamnesis (OR 1.49, 95% confidence interval (CI), 1.63–3.67; *p* = 0.0006). These results were also seen when comparing the right (OR 1.8, CI 1.37–3.17; *p* = 0.0034) and the left lung (OR 1.3, CI 0.74–1.73; *p* = 0.002,) between both groups of patients (Table 1).

### 4.4. LUS Correlation with Symptoms of Previous COVID-19 Infection

Of the total 123 patients in group 2 with a previous COVID-19 infection, 47 (38.21%) were asymptomatic, while 76 (61.79%) had symptoms ranging from mild to moderate. In the symptomatic subgroup, 56 (45.53%) had symptoms classified as mild, whilst the remaining 20 (16.26%) had moderate symptomatology. Globally, children with a symptomatic COVID-19 infection showed higher total value of global LUS score than asymptomatic patients (*p* = 0.0228) (Table 3 and Table A2).

When comparing LUS scores between asymptomatic patients and patients with mild symptomatology, no difference was observed. This observation was made globally (*p* = 0.113) and in the right (*p* = 0.69) and in the left (*p* = 0.06) lung. Contrarily, asymptomatic patients had significantly lower global LUS scores when compared with patients with moderate symptomatology (*p* = 0.0025) and was even confirmed when the right (*p* = 0.026) and left (*p* = 0.04) lung were analyzed individually (Table 3).

Surprisingly, after 1:1 matching by BSA, no differences in LUS score were evaluated between patients with an absence of COVID-19 infection and those who were asymptomatic (right LUS *p* = 0.247; left LUS *p* = 0.919). Inversely, patients with mild and moderate symptoms had a higher right and left LUS score than the group without COVID-19 infection (*p*-value ranging from *p* < 0.0001 to 0.048) (Table 3).

### 4.5. Confounders: Differences among Age Groups

Four age groups were evaluated, namely age group 1 (31 days–≤ 24 months), age group 2 (2–≤5 years), age group 3 (5–≤11 years), and age group 4 (11–≤ 18 years), and we observed no significant differences in global, right, or left LUS score (Table A3).

Inter-observer and intra-observer CV and ICC showed good reproducibility (Table A4).

## 5. Discussion

LUS is widely implemented for the diagnosis of pulmonary disease in children [1,2,3,4,5,6], and the COVID-19 pandemic has shown its utility in daily practice [9,10,11]. Despite its utility, systems for pulmonary disease severity classification by LUS in a pediatric population are lacking [1,2,4], and the definition of a normal LUS examination remains unclear. Isolated artifacts at LUS examination (e.g., B-line, short subpleural vertical artifacts) are usually considered “physiological”, further complicating the difficult differentiation between normal and pathological findings. Additionally, the difference between normal and pathological findings has never been completely defined. Thus, the diagnostic power of LUS in differentiating between healthy and mild pulmonary infections may be limited. Often, COVID-19 infection is mildly symptomatic or asymptomatic in children [8,9,10,11]. In this context, LUS might be able to help clinicians understand whether mild pulmonary involvement is present in this specific population. In this study, we report data with normal LUS examinations in a large population of healthy children. We also compared LUS findings in healthy children without previous COVID-19 infection and in those who recovered from COVID-19 infection.

Our data demonstrated that artifacts are present in almost all children. These findings include short subpleural vertical artifacts and rare B-lines typically seen at the base, apically, and near the spine. Artifacts may be present either alone or in combinations and may be limited to the extremities (base and apex of the lung) or diffuse through the entire lungs. In healthy children without COVID-19 infection, artifacts were usually limited to short subpleural vertical artifacts that may be seen only in posterior basal segments or extended to paraspinal and apical areas. In addition, B-lines could also be observed.

Various ways to quantify lung congestion and pulmonary disease are currently available [15,16,17,18,19,20,21]. In adults, scores for classification of lung congestion in heart failure sum B-lines in each scanning area where more than three B-lines are detached [20]. According to the sum of B-lines generated, lung disease is classified into four categories (none, mild, moderate, and severe) [18,19]. In children, simplified semiquantitative [15,17] or qualitative [9,16,20] scores are generally used.

The score of zero, indicating no lung disease, is characterized by the absence of significant B-lines, which corresponds to a maximum of two B-lines, for each scanning area [15,20,21,22]. Meanwhile, pulmonary disease is graded in an increasing order of severity from grade 1 (separated B-lines) to grade 2 (coalescent B-lines) and up to a final grade 3 (complete loss of aeration or lung consolidation).

In our experience, however, more than two B-lines for each scanning area were frequently observed in otherwise normal lungs. Thus, we accepted a maximum of four B-lines as the score of zero (normal).

Notably, the presence of short subpleural vertical artifacts is not considered in these kinds of classifications, and the presence of zero or two B-lines is scored equally. Yet, short subpleural vertical artifacts are also often present in normal examinations. Additionally, the presence of B-lines itself, especially if seen in multiple scanning areas, may generate confusion in inexperienced operators.

Thus, our data help to clarify how often these minor findings are encountered in a normal LUS examination and where they are more commonly located within the lung, further granting the possibility to distinguish what is normal from pathological.

This study also provides further innovative elements through the description of LUS performance in children who recovered from recent (within 2 months), not severe COVID-19 infection. Additionally, our data show that LUS artifacts are more pronounced in healthy children who experienced a recent, not severe COVID-19 infection. LUS findings were similar among children having asymptomatic COVID-19 disease and those who experienced mild symptoms, while LUS artifacts were more pronounced in those having a moderate COVID-19 illness. None of the children we examined had a previous severe or critical COVID-19 infection. LUS has been extensively used for the diagnosis and follow-up of COVID-19 pulmonary disease in adults [6,7,23]. Applications in children with COVID-19 have been also reported, but data are scarce, fragmentary, and limited to acute care settings [9,10,11].

In 2020, Gregori et al. [9] retrospectively analyzed the data of 32 children (age range 0.5–14.67 years) who presented with persistent cough for a minimum of 3 days and with suspected COVID-19. They demonstrated the presence of multiple B-lines with or without thickening of the pleural line in sixteen cases: one case had coalescent B-lines, and in three patients, lung consolidation was observed. Interestingly, they did not observe a significant relationship between the degree of pulmonary commitment assessed at the US level and the clinical symptoms of the patient [9]. An important remark is that the authors were unaware of actual presence of COVID-19 in most cases since nasal swab was only performed in ten cases, and COVID-19 was only detected in two patients [9].

Sainz et al. [10] prospectively included 20 children (median age 5.2 years; IQR: 2.9–11.4 years) that presented with symptoms attributable to COVID-19 infection at the Pediatric Department of a tertiary hospital in Madrid. Presence of COVID-19 was confirmed in half of the cases. Here, LUS examination was negative in three cases with COVID-19, while in the seven remaining children, LUS abnormalities were noted bilaterally (75%). These abnormalities consisted of pleural irregularities (50% anterior, 70% posterior), more than three B-lines per intercostal space uni- or bilaterally (70%), and consolidations (30%) [10].

To the best of our knowledge, there are no reports comparing LUS findings in children who experienced an asymptomatic or complicated with mild-to-moderate symptoms of COVID-19 infection in the outpatient setting. Thus, our data may be of relevance to understand whether COVID-19 is associated with pulmonary involvement either in an asymptomatic population or those with a mild symptomatic presentation. Our findings suggest that that mild pulmonary involvement may be present even in those presenting with mild symptomatic COVID-19 infections. Fortunately, signs of significant pulmonary disease were absent in all patients, including patients presenting with moderate symptoms.

## 6. Limitations

The present study has limitations. We evaluated only the Caucasian ethnic group; however, this eliminated bias due to racial diversity. An intrinsic limitation is the lack of standardized scoring systems to classify normal and mild disease at LUS examination. To overcome this limitation, we employed a novel score. Another possible limitation is the variation in settings of the probe and ultrasound machine between observations. In our study, we used a fixed setting, as indicated in international recommendations [2]. Furthermore, a serological test to detect previous COVID-19 infections was not performed routinely. Thus, a percentage of patients included in group 1 (patients with absence of COVID-19 infection) might have undergone an asymptomatic COVID-19 infection, which might introduce a limited amount of bias into our analysis. To combat this, we screened all included children for school admission to detect any absence at school, which might point to a possible COVID-19 infection. Additionally, the choice of a 2-week negativization period to include patients with prior COVID-19 infection may be too limited since multisystem inflammatory syndrome in children (MIS-C) usually presents 1 month after COVID-19 infection [23,24,25]. However, none of our included patients developed MIS-C in the period after our study. Lastly, we had a too-limited sample size of children with severe COVID-19 infection to be analyzed. Our data, however, may serve as baseline for the study of long-term pulmonary consequence in these patients.

## 7. Conclusions

In healthy Caucasian children, we report a pattern of normality on lung ultrasound screening. In addition, we report LUS patterns in children who experienced a recent, not severe COVID-19 infection. Our data reveal that a complete absence of the artifacts is rare and that short subpleural vertical artifacts and rare B-lines are almost universally observed posteriorly at the base, the apex, and near the spine. However, a higher rate of these “physiological” LUS artifacts was detected in children who experienced a COVID-19 infection, with more pronounced artifacts in those having moderate COVID-19 illness. This information may serve as a baseline for lung evaluation via LUS in children with suspected pulmonary disease and might be helpful to understand the value of LUS in the follow-up of children with a confirmed COVID-19 infection.

## Figures and Tables

**Figure 1 jcm-11-05999-f001:**
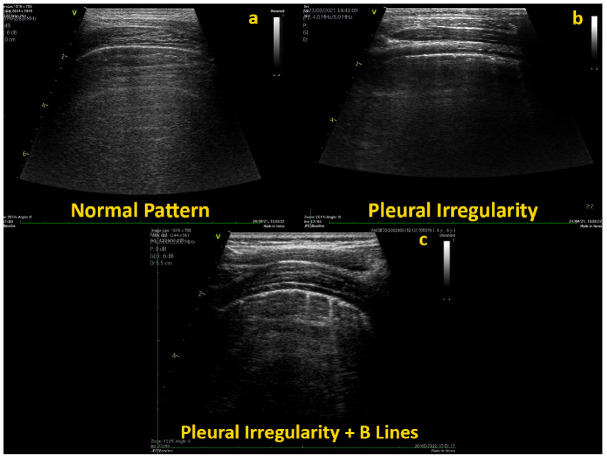
Normal LUS examination with: (**a**) no B-lines and no subpleural artifacts; (**b**) mild sub-pleural artifacts and no B-lines; and (**c**) mild subpleural artifacts and rare B-lines.

**Figure 2 jcm-11-05999-f002:**
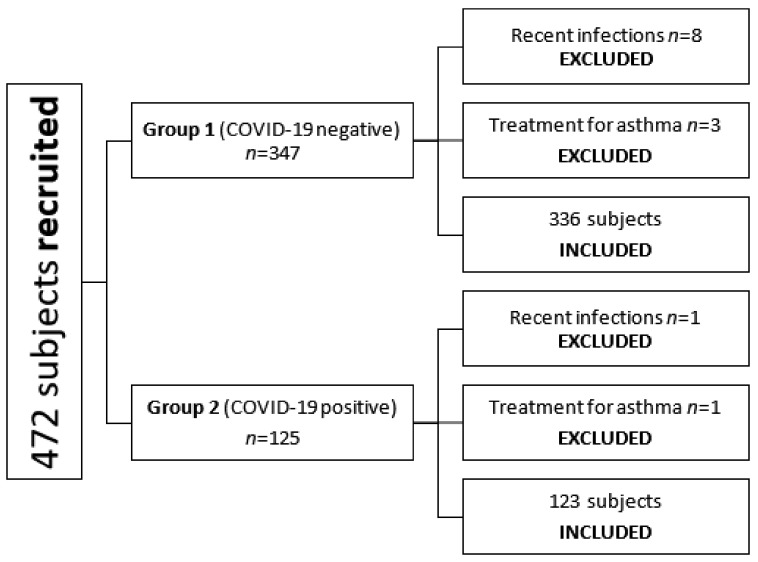
Patient selection.

**Table 1 jcm-11-05999-t001:** Demographic and LUS data of enrolled patients.

	Whole Population *n* = 459					Group 1 (Negative) *n* = 336					Group 2 (Positive) *n* = 123					Group 1 vs. Group 2
	Mean	SD	Min	Max	Median	Mean	SD	Min	Max	Median	Mean	SD	Min	Max	Median	*p*
**Age (years)**	10.564	3.839	0.293	17.868	10.701	10.498	3.941	0.293	17.868	10.618	10.745	3.555	0.934	17.847	11.118	0.543
**Weight (kg)**	39.556	17.11	8.000	133.00	37.000	39.275	17.30	8.000	133.00	36.500	40.321	16.65	13.00	83.000	39.000	0.563
**Height (cm)**	141.56	23.80	11.00	189.00	144.00	140.77	24.47	11.00	187.00	144.00	143.71	21.82	80.00	189.00	145.00	0.242
**BSA (m^2^)**	1.231	0.355	0.339	2.051	1.216	1.222	0.355	0.339	2.051	1.202	1.255	0.353	0.548	2.038	1.255	0.379
**Right LUS**	0.268	0.176	0.0	1.0	0.2	0.254	0.16	0.0	0.8	0.2	0.308	0.206	0.0	1.0	0.3	0.0034
**Left LUS**	0.274	0.178	0.0	1.0	0.2	0.262	0.168	0.0	0.8	0.2	0.306	0.200	0.0	1.0	0.2	0.02
**Global LUS**	0.271	0.163	0.0	1.0	0.2	0.259	0.154	0.0	0.8	0.2	0.306	0.185	0.0	1.0	0.25	0.006

BSA, body surface area; LUS, lung ultrasound.

**Table 2 jcm-11-05999-t002:** Pattern distribution and classification.

Total *n* = 459			Group 1 (Negative) *n* = 336		Group 2 (Positive) *n* = 123	
	Total LUS Score		Total LUS Score		Total LUS Score	
	*n*	%	*n*	%	*n*	%
Pattern 1	41	8.93	35	10.42	7	5.69
Pattern 2	26	5.66	16	4.76	10	8.13
Pattern 3	206	44.88	171	50.89	37	30.08
Pattern 4	117	25.49	76	22.62	46	37.4
Pattern 5	41	8.93	22	6.55	12	9.76
Pattern 6	17	3.7	10	2.98	6	4.88
Pattern 7	11	2.4	6	1.79	5	4.07

**Table 3 jcm-11-05999-t003:** Demographic and global LUS data of patients who experienced COVID-19 sorted by symptoms.

	Asymptomatic COVID-19 *n* = 47			Mild Symptomatology *n* = 56			Moderate Symptomatology *n* = 20			Asympt vs. Mild	Asympt vs. Mod	Mild vs. Mod	Neg vs. Asympt	Neg vs. Mild	Neg vs. Mod
	Mean	SD	Min– Max	Mean	SD	Min– Max	Mean	SD	Min– Max	*p*	*p*	*p*	*p*	*p*	*p*
Age (y)	10.797	3.943	0.934– 17.847	10.311	3.214	1.477– 15.529	12.048	3.336	6.762– 17.479	0.487	0.247	0.06	0.4	0.213	0.09
Weight (kg)	41.93	18.25	16.7– 83.0	36.74	14.15	13.0– 75.0	47.68	17.43	24.0– 78.0	0.103	0.262	0.01	0.12	0.11	0.08
Height (cm)	145.63	22.355	107.0– 189.0	140.04	21.78	80.0– 175.0	150.53	19.07	120.0– 180.0	0.195	0.423	0.078	0.049	0.07	0.32
BSA	1.286	0.383	0.712– 2.038	1.184	0.315	0.548– 1.863	1.402	0.349	0.912– 1.949	0.133	0.278	0.02	0.74	0.62	0.13
Global LUS	0.26	0.15	0.1– 0.8	0.312	0.19	0.0– 1.0	0.392	0.23	0.1– 1.0	0.19	0.01	0.14	0.7	0.018	0.0004

After 1:1 matching by BSA. Asympt, asymptomatic; Mod, moderate symptomatology; Neg, negative; SD, standard deviation; BSA, body surface area; LUS, lung ultrasound.

## Data Availability

The data presented in this study are available on request from the corresponding author.

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
