# Peer review of "Lung Ultrasound Findings in Healthy Children and in Those Who Had Recent, Not Severe COVID-19 Infection"

_jcm, 2022, doi:10.3390/jcm11205999_

Round 1

Reviewer 1 Report

Thank you very much for giving me a great opportunity to read this article. This study consisted of two main parts; (1) establishing lung ultrasound assessment, (2) investigating the LUS difference between children with and without a history of COVID-19. The authors presented a scoring system of children's LUS and indicated that those who had recovered from COVID-19 may have more pronounced imaging findings. This is highly relevant and interesting topic since, in adults, it is already known that COVID-19 disease severity correlates with residual CT abnormalities and lung function tests. However, this study has major limitations to be addressed. 

Major:

1. I am not quite sure this combined design of validation and investigational study is appropriate. The authors gathered data of the healthy children and established a scoring system to assess LUS. These findings were deemed "normal" assuming that the COVID-19 subjects should be "abnormal." However, nobody knows if children who recovered from COVID-19 present abnormal findings in LUS (if it is already known, the novelty of this manuscript is substantially jeopardized). I personally think it is more scientific to validate the scoring system first and adopting that into the investigation.

2. Although the authors quantified the variability in LUS observation to guarantee the reproducibility, it still remains a major concern that one of the observers (MC) also reviewed the stored images from the standpoint of blinding. There could have been a serious recall bias or examiner's bias if the reviewer already knew the COVID-19 status of the examinees. 

3. Where are the data of vaccinated children? Without those data, this manuscript does not support the authors' conclusion "we report LUS patterns in both asymptomatic children who experienced a COVID-19 infection and in children who were vaccinated against COVID-19."

Minor:

1. Are there any particular reasons why the authors only included Caucasian children? This limits the external validity of this study.

2. Please include the effect of measures (Odds ratio, relative risk reduction, etc) in addition to p-value. Only reporting p-value does not provide any clinical implication; therefore it is no longer recommended.

3. In adults, it is known that despite the residual imaging abnormalities in patients who recovered from severe COVID-19, it gradually resolves. In this study, were there any data available when it comes to the time period after the COVID-19 recovery in each participants?

Caucasian

only p value

Author Response

Comments and Suggestions for Authors

Thank you very much for giving me a great opportunity to read this article. This study consisted of two main parts; (1) establishing lung ultrasound assessment, (2) investigating the LUS difference between children with and without a history of COVID-19. The authors presented a scoring system of children's LUS and indicated that those who had recovered from COVID-19 may have more pronounced imaging findings. This is highly relevant and interesting topic since, in adults, it is already known that COVID-19 disease severity correlates with residual CT abnormalities and lung function tests. However, this study has major limitations to be addressed.

Major:

  1. I am not quite sure this combined design of validation and investigational study is appropriate. The authors gathered data of the healthy children and established a scoring system to assess LUS. These findings were deemed "normal" assuming that the COVID-19 subjects should be "abnormal." However, nobody knows if children who recovered from COVID-19 present abnormal findings in LUS (if it is already known, the novelty of this manuscript is substantially jeopardized). I personally think it is more scientific to validate the scoring system first and adopting that into the investigation.

Response: we did not assume first that children who experienced Covid 19 had abnormal findings. Our aim was to compare findings of children who did not experienced LUS with those who had LUS

  1. Although the authors quantified the variability in LUS observation to guarantee the reproducibility, it still remains a major concern that one of the observers (MC) also reviewed the stored images from the standpoint of blinding. There could have been a serious recall bias or examiner's bias if the reviewer already knew the COVID-19 status of the examinees.

Response:  We apologize with the reviewer since there was a mistake in the text- Two expert operators (E.F., and P.M.) perform all the examination and another expert operator (M.C.) performed off-line blinded analysis. We have corrected within the method section

  1. Where are the data of vaccinated children? Without those data, this manuscript does not support the authors' conclusion "we report LUS patterns in both asymptomatic children who experienced a COVID-19 infection and in children who were vaccinated against COVID-19."

Response: again, we apologize with the reviewer, it was a refuse, from a previous version of the text

Minor:

  1. Are there any particular reasons why the authors only included Caucasian children? This limits the external validity of this study.

Response: in our area the 95% of children in outpatient are Caucasian

  1. Please include the effect of measures (Odds ratio, relative risk reduction, etc) in addition to p-value. Only reporting p-value does not provide any clinical implication; therefore it is no longer recommended.

Response: we added data requested by the reviewer where possible

  1. In adults, it is known that despite the residual imaging abnormalities in patients who recovered from severe COVID-19, it gradually resolves. In this study, were there any data available when it comes to the time period after the COVID-19 recovery in each participants?

Response: we detail in the text and the title that we studied children with recent not severe Covid infections.

Reviewer 2 Report

The article ,Lung Ultrasound Findings in Healthy Children and in Those Who Had COVID-19: Does COVID-19 Leave Pulmonary Damage in Children?' gives insights on usage of lung ultrasound in children specifically in the case of previous COVID-19 infection. The article is well written and easy to read. The following comments may aid to improve the manuscript. 

- Title: I propose to adjust the title of the article. Whether COVID-19 leaves lung injury cannot be inferred from the data. Rather, as referred to in the conclusion, the article gives insights on results of LUS screening in healthy children, those with  aymptomatic and those with symptomatic COVID-19 disease.  

- Abstract: the Abstract misses a ,results' and ,conclusion' section

- In the methods section, the authors indicate, that COVID-19 symptoms are classified using the treatment guidelines classification. Were only children with mild to moderate symptoms included and none with severe or critical infection? The reader should be informed if no patients with more than moderate symptoms presented at hospital or if these were excluded from analysis. 

- In Table 2, the ultrasound findings are presented classified by pattern. It is not clear in the manuscript why a distinction between right and left lung is made. For clarity and to improve readability, I would suggest to present the results of both lungs together (the other tables can be moved into the supplement of the article). Otherwise -if there is a pathophysiological rationale behind different results between left and right lung- this should be adressed in the discussion. Furthermore, the p-values comparing group 1 and 2 should be given, escpecially because in the discussion, the authors state, that LUS artifacts are more pronounced in healthy children who have suffered a COVID-19 infection.  

- A significant difference of ultrasound findings in mild versus moderately symptomatic patients is shown in the area of the right lung whilst no significant differences have been demonstrated in the area of the left lung (see Table 3). If the authors insist on keeping a differentiation between both lung fields albeit its difficult readability, this point should be addressed and explained in the discussion.

 - Discussion: Line 234 page 11: Did you mean 'mildly symptomatic' opposed to 'mildy asymptomatic'?

- Discussion: In the last sentense the authors indicate that 'lung ultrasound helps to monitor long-term sequelae of COVID-19'. By contrast, the authors state that signs of significant lung injury were absent in all analyzed patients. Moreover, there is no information about the clinical presentation of the children at time of LUS. This statement should be explained in more detail or deleted if appropriate. 

- In the method section, the score is introduced as a score with predictive value for ICU stays after pediatric cardiac surgery. In the present work, no endpoint is defined, but a correlation to previous corona infection is investigated. The article would be of more value if clinical relevance of ultrasound results was referred to in more detail. 

Author Response

Comments and Suggestions for Authors

The article ,Lung Ultrasound Findings in Healthy Children and in Those Who Had COVID-19: Does COVID-19 Leave Pulmonary Damage in Children?' gives insights on usage of lung ultrasound in children specifically in the case of previous COVID-19 infection. The article is well written and easy to read. The following comments may aid to improve the manuscript. 

- Title: I propose to adjust the title of the article. Whether COVID-19 leaves lung injury cannot be inferred from the data. Rather, as referred to in the conclusion, the article gives insights on results of LUS screening in healthy children, those with  aymptomatic and those with symptomatic COVID-19 disease.  

Response: we have changed the title as suggested by the revieewer

- Abstract: the Abstract misses a, results' and ,conclusion' section

Response: we added results and conclusions section in the abstract. We apologize for the mistake

- In the methods section, the authors indicate, that COVID-19 symptoms are classified using the treatment guidelines classification. Were only children with mild to moderate symptoms included and none with severe or critical infection? The reader should be informed if no patients with more than moderate symptoms presented at hospital or if these were excluded from analysis.

Response: we studied only outpatient children who had only mild to moderate infections-We have detailed in the text, that we have a too limited sample size of children with a previous severe infection to be analyzed with a sufficient statistical power 

- In Table 2, the ultrasound findings are presented classified by pattern. It is not clear in the manuscript why a distinction between right and left lung is made. For clarity and to improve readability, I would suggest to present the results of both lungs together (the other tables can be moved into the supplement of the article). Otherwise -if there is a pathophysiological rationale behind different results between left and right lung- this should be adressed in the discussion. Furthermore, the p-values comparing group 1 and 2 should be given, escpecially because in the discussion, the authors state, that LUS artifacts are more pronounced in healthy children who have suffered a COVID-19 infection.  

- A significant difference of ultrasound findings in mild versus moderately symptomatic patients is shown in the area of the right lung whilst no significant differences have been demonstrated in the area of the left lung (see Table 3). If the authors insist on keeping a differentiation between both lung fields albeit its difficult readability, this point should be addressed and explained in the discussion.

Response: _ differences among the two lungs were minimal. We substitute the table with a single table and moved the tables with details of both lungs in the supplemental material. We also add P as quested by the reviewer

 - Discussion: Line 234 page 11: Did you mean 'mildly symptomatic' opposed to 'mildy asymptomatic'?

Response: we apologize for the error. We corrected.

- Discussion: In the last sentence the authors indicate that 'lung ultrasound helps to monitor long-term sequelae of COVID-19'. By contrast, the authors state that signs of significant lung injury were absent in all analyzed patients. Moreover, there is no information about the clinical presentation of the children at time of LUS. This statement should be explained in more detail or deleted if appropriate.

Response: we eliminated this confusing section as suggested by the reviewer 

- In the method section, the score is introduced as a score with predictive value for ICU stays after pediatric cardiac surgery. In the present work, no endpoint is defined, but a correlation to previous corona infection is investigated. The article would be of more value if clinical relevance of ultrasound results was referred to in more detail. 

Response: we apologize for the confusion generating in the method section, we have tried to better clarify the scoring system

Round 2

Reviewer 1 Report

Thank you very much again for the opportunity to review this article. I appreciated the authors' responses to each comment. 

Overall, the manuscript quality was improved with a more appropriate design and relevant discussion/conclusion.

I would appreciate it if the authors use a language editing service to reduce redundancy and raise the clarity of the manuscript. It is quite often difficult to follow.

In addition, the effect size should be presented more appropriately.

e.g., OR, XX (95% confidence interval [CI], XX-XX; p=.XX).

Just putting OR after the p-value is not scientific or informative.

Author Response

Thank you very much again for the opportunity to review this article. I appreciated the authors' responses to each comment. 

Overall, the manuscript quality was improved with a more appropriate design and relevant discussion/conclusion.

I would appreciate it if the authors use a language editing service to reduce redundancy and raise the clarity of the manuscript. It is quite often difficult to follow.

In addition, the effect size should be presented more appropriately.

e.g., OR, XX (95% confidence interval [CI], XX-XX; p=.XX).

Just putting OR after the p-value is not scientific or informative.

Response: the text has been reviewed by an English native. We perform and extensive edits on tables and sub-headings. The effect size has been presented as requested.

Corrections are in blue in the text